# Volatile Constituents of Some Myrtaceous Edible and Medicinal Fruits from the Brazilian Amazon

**DOI:** 10.3390/foods13101490

**Published:** 2024-05-11

**Authors:** Adenilson S. Barroso, Lais T. Massing, Chieno Suemitsu, Rosa Helena V. Mourão, Pablo Luis B. Figueiredo, José Guilherme S. Maia

**Affiliations:** 1Laboratório de Bioprospecção e Biologia Experimental, Universidade Federal do Oeste do Pará, Santarém 68035-110, PA, Brazil; adenilson.barroso@yahoo.com.br (A.S.B.); laismassing@hotmail.com (L.T.M.); chieno.suemitsu@gmail.com (C.S.); mouraorhv@yahoo.com.br (R.H.V.M.); 2Programa de Pós-Graduação em Ciências Farmacêuticas, Universidade Federal do Pará, Belém 66075-110, PA, Brazil

**Keywords:** edible Brazilian Amazon fruits, volatile concentrates, terpenes and fatty acid derivatives

## Abstract

Native and exotic fruits from the Amazon have varied characteristics, with aroma being a decisive factor in their acceptance for medicinal use as a nutraceutical supplement. This work aimed to analyze the chemical constituents of the volatile concentrates of some Myrtaceous fruit species sampled in the Brazilian Amazon. The fruit’s pulps were subjected to simultaneous distillation–extraction, and gas chromatography–mass spectrometry was used to analyze their volatile chemical composition. In the volatile concentrate of *Eugenia stipitata* (Araçá-boi) α-pinene (17.5%), citronellyl butanoate (15.6%), and pogostol (13.5%) were identified as primary constituents; *Eugenia uniflora* (Ginja) concentrate comprised curzerene (30.5%), germacrone (15.4%), atractylone (13.1%), and (*E*)-β-ocimene (11.1%); in *Myrciaria dubia* (Camu-Camu), α-pinene (55.8%), (*E*)-β-ocimene (13.1%), and α-terpineol (10.0%) were present; in *Psidium guajava* (Goiaba) were (2*E*)-hexenal (21.7%), hexanal (15.4%), caryophylla-4(12),8(13)-dien-5-β-ol (10.5%), caryophyllene oxide (9.2%), and pogostol (8.3%); and in *Psidium guineense* (Araçá), limonene (25.2%), ethyl butanoate (12.1%), epi-β-bisabolol (9.8%), and α-pinene (9.2%) were the main constituents. The analyzed volatile concentrates of these fruit species presented a significant diversity of constituents with a predominance of functional groups, such as monoterpenes, sesquiterpenes, and fatty acid derivatives, originating from the plant’s secondary metabolism and playing an important role in their nutritional and medicinal uses.

## 1. Introduction

The Amazon region is the last stronghold of potentially valuable plants awaiting domestication and economic exploitation. The use of exotic fruits that have been introduced worldwide, such as apples and oranges, has undergone improvement across centuries in a continuous process, the memory of whose initiation has already been lost to time. Humanity began to domesticate plants around ten thousand years ago, while the history of the domestication of Amazonian fruits has only begun to be written [1].

Several fruit species native to the Amazon have been commercialized and consumed for medicinal and nutritional purposes. They are known for having a pleasant flavor and juicy pulp, representing significant economic potential; however, they still require domestication and genetic improvement studies. These studies must be resumed urgently due to increases in deforestation, which introduces a high risk of species extinction, in addition to the fact that some of these fruits still need to be adequately studied and scientifically classified [2,3,4,5,6,7].

A significant variety of fruits are consumed in Brazil, representing one of the primary sources of vitamins, minerals, fiber, aromas, and antioxidants in the diet of the native population. In this context, Brazil is considered one of the great centers of global biodiversity, and has many tropical fruits with different and pleasant flavors [2]. Due to its territorial extension, geographical position, climate, and soil, Brazil produces fruits in tropical, subtropical, and temperate areas, being the third largest fruit producer in the world, after China and India, with 42 million tons produced per year and more than 2.2 million hectares of fruit crops planted across the country. More than 30% of fresh fruits produced in Brazil are exported to different parts of the world [3,4].

Given this context, the region of the Middle Amazon River, spanning the State of Pará and made up of the municipalities of Óbidos, Juruti, Oriximiná, Terra Santa, and Santarém, has made available several species of native fruits, some produced on a large scale and sold for different purposes, in addition to others fruits unnoticed by consumers, resulting from limited stocks and logistics (IBGE, 2019) [8]. In addition to nutritional value, other attributes existing in native fruits and those cultivated in this region of the Middle Amazon River are their outstanding aromas, composed of a significant diversity of different volatile constituents which, despite representing a low percentage of the total mass of the fruit (around 0.05% to 1.0%), contribute to the taste, flavor, and acceptability of these fruits. Furthermore, scientific knowledge of the chemical constituents responsible for the characteristic aromas of tropical fruits is justified by the importance they can play in the quality of their products. The attractive tropical fruit flavor has stimulated growing consumer interest around the world. In this context, the Amazon stands out for its outstanding natural diversity of fruits, with characteristic flavors that require the identification of their volatile constituents, also representing a promising area for research into the typical aromas of this region [7].

In fruits, the biosynthetic routes for forming volatile constituents involve enzymatic reactions, producing volatile components such as terpenes, sulfur compounds, derivatives of fatty acids, derivatives of amino acids, and those originating from fermentation. The enzymatic generation of volatile constituents derived from fatty acids is one of the main routes that leads to the formation of the characteristic flavor of fruits. As reactions occur, the aroma of the fruit can change, and aldehydes and ketones, for example, can be converted into the corresponding alcohols, presenting more prominent aromas [9].

The present work aimed to characterize the chemical composition of the volatile concentrates of the Myrtaceae edible fruits, including *Eugenia stipitata* (Araçá-boi), *Eugenia uniflora* (Ginja), *Myrciaria dubia* (Camu-Camu), *Psidium guajava* (Goiaba), *Psidium guineense* (Araçá), which are sold in fairs and markets in the cities of Santarém, Juruti, Oriximiná, and Terra Santa, in the Lower Amazon River region, Brazil (see Figure 1).

## 2. Materials and Methods

### 2.1. Plant Material

The fruits *Eugenia stipitata* (Araçá-boi), *E. uniflora* (Ginja), *Myrciaria dubia* (Camu-Camu), *Psidium guajava* (Goiaba), and *P. guineense* (Araçá) (Figures 2, 4, 6, 8 and 10), which provided their pulps for this work, were obtained in fairs and markets in the cities of Santarém (Coordinates: 02°26′35″ S/54°42′30″ W), Juruti (Coordinates: 02°09′08″ S/56°05′32″ W), Oriximiná (Coordinates: 01°45′56″ S/55°51′58″ W), and Terra Santa (Coordinates: 02°06′15″ S/56°29′13″ W), located in the Lower Amazon River region, Pará state, Brazil (see Figure 1).

The selection of fruits was made considering their seasonality and maturity, integral characteristics, natural shape without deformations, and absence of microbiological contamination. The fruits were washed in running water, measured, and weighed, and their pulp (edible part) was processed to remove seeds and skins. Then, the fruit pulps were frozen for subsequent analysis.

### 2.2. Obtaining and Analyzing Volatile Concentrates

The fruit pulps were subjected to microdistillation–extraction in a Likens and Nickerson-type apparatus (30 g in total, 2 h, duplicate) to obtain their volatile concentrates, using *n*-pentane (99% HPLC grade, 3 mL) as the solvent [10].

The volatile concentrates were subjected to GC and GC–MS analysis. This was performed on a GCMS-QP2010 Ultra system (Shimadzu Corporation, Tokyo, Japan), equipped with an AOC-20i auto-injector and the GCMS-Solution software (Shimadzu, Japan) containing the Adams (2007) and Mondello (2011) libraries [11,12]. An Rxi-5ms (30 m × 0.25 mm; 0.25 μm film thickness) silica capillary column (Restek Corporation, Bellefonte, PA, USA) was used. The conditions of analysis were as follows: injector temperature, 250 °C; oven temperature programming, 60–240 °C (3 °C min^−1^); helium as the carrier gas, adjusted to a linear velocity of 36.5 cm s^−1^ (1.0 mL min^−1^); split mode injection (split ratio 1:20) of 1.0–2.0 µL of the *n*-pentane solution; electron ionization at 70 eV; and ionization source and transfer line temperatures of 200 and 250 °C, respectively. The mass spectra were obtained by automatically scanning every 0.3 s, with mass fragments in the range of 35–400 *m*/*z*. The retention index was calculated for all volatile components using a homologous series of C8–C40 *n*-alkanes (Sigma-Aldrich, Milwaukee, WI, USA) according to the linear equation of van den Dool and Kratz (1963) [13]. Individual components were identified by comparing their retention indices and mass spectra (molecular mass and fragmentation pattern) with those existing in the GCMS-Solution system libraries [11,12]. The quantitative data regarding the volatile constituents were obtained using a GC2010 Series gas chromatograph, operated under conditions similar to the GC–MS system. The relative amounts of individual components were calculated by peak-area normalization using a flame ionization detector (GC-FID). Chromatographic analyses were performed in duplicate.

### 2.3. Multivariate Statistical Analysis

Principal component analysis (PCA) was applied to verify the interrelationship of the samples of volatile concentrates analyzed with the classes of identified compounds, including monoterpene hydrocarbons (MH), oxygenated monoterpenes (OM), sesquiterpene hydrocarbons (SH), oxygenated sesquiterpenes (OS), benzenoids/phenylpropanoids (BP), and fatty acid derivatives (FA). The data matrix was standardized for multivariate analysis by subtracting the mean and dividing it by the standard deviation. Hierarchical cluster analysis (HCA), considering the Euclidean distance and complete linkage, was used to verify the similarity of the samples based on the distribution of the constituents selected in the PCA analysis (Software Minitab, free version 390, Minitab Inc., State College, PA, USA) [14].

## 3. Results and Discussion

### 3.1. Eugenia stipitata McVaugh–Myrtaceae

**Botanical description:** It is an ornamental leafy tree or shrub known as Araçá-boi. It is 3.0–15.0 m tall, densely branched, and lacks apical dominance. The stem is brown to reddish-brown, the bark is flaky, and young branches are covered with short, velvety, brown hairs that are lost with age. Leaves are arranged in an opposing formation, and are simple and without stipule. The petiole is short, at 3 mm long; the leaf blade is ovate to somewhat broadly elliptic, and is 8–19 cm long and 3.5–9.5 cm wide. The apex of the leaves acuminates, while the base is rounded and often subcordate. The leaf margins are entire, and the leaves are dull in color; they are dark green on top, with 6–10 pairs of impressed lateral veins, and pale green, shortly pilose, with scattered hairs underneath. The inflorescence racemose pedicles are long, and the bracteoles are linear and 1–2 mm long. The calyx lobes are rounded, broader than long, and overlap in the bud. There are five petals, which are white in color, obovate and ciliate, and 7–10 mm long and 4 mm wide. There are about 70 stamens of 6 mm in length. There are four locules, each locule containing 5–8 ovules that are 5–8 mm long. The fruits are oblate or spherical berries measuring 2–10 × 2–12 cm, weighing 50–750 g. They are light green at first, and turn pale or orange-yellow when ripe. They are soft, with a thin, velvety skin enclosing a juicy, thick pulp that accounts for as much as 60% of the fresh fruit. There are approximately 12 seeds in each fruit (see Figure 2) [15]. They fruit from November to May in all Amazon regions. The pleasant-tasting Araçá-boi fruit is rich in vitamins A, B1, and C, and it is used in soft drinks, juices, ice creams, and sweets.

**Synonimy:** *Eugenia stipitata* subsp. *stipitata* McVaugh, *E. stipitata* subsp. *sororia* McVaugh [5].

**Geographic distribution:** It is a fruit tree native to the Peruvian Amazon. It is found in the wild in many areas of the region, and it has proliferated across the Ucaiali River basin in Peru. In the state of Amazonas, Brazil, it is cultivated on a domestic scale by the caboclo and indigenous populations of the Solimões River [5]. See Table 1 concerning the volatile constituents identified in *Eugenia stipitata*.

Monoterpenes hydrocarbons (28.5%), oxygenated monoterpenes (25.5%), and oxygenated sesquiterpenes (20.9%) predominated in the volatile concentrate of *E. stipitata*, followed by sesquiterpene hydrocarbons (13.1%) and fatty acids and derivatives (10.8%). The main constituents were α-pinene (17.4%), citronellyl butanoate (15.6%), pogostol (13.5%), α-terpineol (9.6%), β-pinene (6.8%), δ-elemene (4.1%), hexyl hexanoate (3.5%), *epi*-α-muurolol (3.2%), and γ-muurolene (2.6%), comprising 76.3% of its volatile concentrate (see Figure 3).

The volatile compositions of the fruits and leaves of *E. stipitata* have been previously reported for samples collected at various locations. A fruit sample collected in Manaus, Brazil, showed germacrene D, β-pinene, and α-pinene as its main constituents [16]; a fruit sample collected in Caquetá, Colombia exhibited ethyl octanoate, ethyl dodecanoate, ethyl decanoate, 1-hexanol, 2-methyl-butanoic acid, hexanoic acid, and octanoic acid, in decreasing order [17]; a leaf sample collected in Azores, Portugal, showed (*E*)-caryophyllene, caryophyllene oxide, and α-pinene as primary compounds [18]; and in a leaf sample collected in the Araripe region, Pernambuco, Brazil, β-eudesmol, γ-eudesmol, elemol, and caryophyllene oxide predominated as the main constituents [19].

### 3.2. Eugenia uniflora L.–Myrtaceae

**Botanical description:** It is a shrub standing at between 1.5 and 8.0 m tall that branches from the base. It is known as Ginja or Pitanga. The leaves are simple, arranged opposite to each other, chartaceous, ovate, 1.5–5.0 m long and 1.0–3.5 m wide, dark green and shiny, and shortly petiolate; they have a rounded base, and a short obtuse-acuminate apex. Flowers are solitary or in groups of 2 to 3, axillary, and with filiform pedicels 2–3 cm long. The corolla of the flower has four white petals, is slightly fragrant, and has numerous stamens. The fruit has an oblate berry 2–3 cm in diameter with 7–10 longitudinal buds, a persistent calyx, and smooth, shiny skin that is red when ripe. The orange pulp is juicy, has a sweet flavor, is a little astringent, and contains 1–2 greenish-white seeds [5] (Figure 4). Fruiting has been observed throughout the year. The Ginja, or Pitanga, fruit has a pleasant flavor and is consumed fresh, in salads, and in the preparation of jellies and ice creams.

**Synonimy:** *Eugenia brasiliana* L. (Aubl.), *E. costata* Cambess, *E. indica* Nicheli, *E. michelii* Lam., *E. microphylla* Barb. Rodr., *Myrtus brasiliana* L., *M. willdenowii* Spreng., *Plinia rubra* L., *Stenocalyx affinis* O. Berg, *S. michelii* (Lam.) O. Berg, *S. uniflorus* (L.) Kausel, *Syzygium michelii* (Lam.) Duthie, among others [20].

**Geographic distribution:** Originally from Brazil, this fruit is now spread throughout South America, the Caribbean islands, Central America, and South Florida. See Table 2 concerning the volatile constituents identified in *Eugenia uniflora*.

The primary compound classes of E. uniflora volatile concentrate were sesquiterpene hydrocarbons (41.3%), oxygenated sesquiterpenes (39.8%), and monoterpene hydrocarbons (17.5%), while its main constituents were curzerene (30.5%), germacrone (15.4%), atractylone (13.1%), (E)-β-ocimene (11.1%), (Z)-β-ocimene (4.6%), and trans-β-elemenone (4.1%), comprising 78.8% of the volatile concentrate (see Figure 5).

The volatile composition of the fruits and leaves of E. uniflora have been previously reported based on samples collected from various locales. A fruit sample collected in Pinar del Rio, Cuba exhibited curzerene, bergaptene, myrcene, (E)-β-ocimene, and limonene as its primary constituents [21]; a fruit sample collected in Pernambuco, Brazil, showed a high content of (E)-β-ocimene, (Z)-β-Ocimene, and β-pinene [22]; a fruit sample collected in Pelotas, Rio Grande do Sul, Brazil, predominately contained hexadecanoic acid, (E)-β-ocimene, α-selinene, and germacrene B [23]; and fruit and leaves samples collected in Ibadan, Nigeria, contained curzerene, selina-1,3,7(11)-trien-8-one, selina-1,3,7(11)-trien-8-one epoxide, atractylone, furanodiene, and germacrone as their major compounds [24]. The essential oils of the leaves and thin branches of E. uniflora cultivated in the city of Belém, Brazil were investigated, and the main components were germacrone, curzerene, and germacrene B (15.6%) [25]; conversely, in the oil of leaves collected in Goiânia, Santo Antonio de Goiás, Nova Veneza e Anápolis, Goías, Brazil, the main constituents were germacrene A, B, and C, atractylone, curzerene, selina-1,3,7(11)-trien-8-one, and selina-1,3,7(11)-trien-8-one epoxide [26].

### 3.3. Myrciaria dubia (Kunth) McVaugh–Myrtaceae

**Botanical description:** Known as Camu-Camu, it is a small shrub typically measuring 1–3 m, but it can reach up to 8 m in height. The leaves are simple and oriented opposite to one another. They are elliptical or broadly ovate in shape, 6–10 cm long and 1.5–3.0 cm wide, and have an obtuse or rounded base, long-acuminate apex, and delicate lateral veins. Axillary inflorescences are formed by subsessile flowers arranged in decussate pairs, and the flowers are white and fragrant. The fruit is a spherical berry measuring 2.0–2.5 cm in diameter, with thin, smooth, shiny skin that is red to blackish-purple in color. It has a slightly pinkish juicy pulp that contains two seeds [5] (Figure 6). This species fruits from November to March in all Amazon regions. The Camu-Camu fruit has an acidic flavor due to its vitamin C content and is used in soft drinks, ice cream, liqueur, jellies, and sweets.

**Synonimy:** *Psidium dubium* Kunth, *Eugenia grandiglandulosa* Kiaersk, *Marlierea macedoi* D. Legrand, *Myrciaria divaricata* (Benth.) O. Berg, *M. lanceolata* O. Berg, *M. obscura* O. Berg, *M. paraensis* O. Berg, *M. phillyraeoides* O. Berg, *M. riedeliana* O. Berg, *M. spruceana* O. Berg, *Myrtus phillyraeoides* (O. Berg) Willd., and *Psidium dubium* Kunth, among others [27].

**Geographic distribution:** This species is distributed northwest of the Brazilian Amazon, and across Peru and Venezuela, in semi-flooded areas. See Table 3 concerning the volatile constituents identified in *Myrciaria dubia*.

Monoterpenes hydrocarbons (79.6%) and oxygenated monoterpenes (11.5%) predominated in the volatile concentrate of *E. stipitata*, followed by sesquiterpenes hydrocarbons (5.2%). The main constituents were α-pinene (55.8%), (*E*)-β-ocimene (13.1%), α-terpineol (10.0%), (*E*)-caryophyllene (4.2%), limonene (3.7%), terpinolene (2.9%), and β-pinene (2.6%), comprising a total of 92.3% of the volatile concentrate (see Figure 7).

Franco and Shibamoto (2000) [16] also identified α-pinene, limonene, and β-caryophyllene as the major constituents of the volatile concentrate of Camu-Camu fruit collected in Manaus, Brazil. Furthermore, Quijano and Pino (2007) [28] highlighted limonene, α-terpineol, and α-pinene as significant components of a volatile concentrate extracted from fruits sampled in Caquetá, Colombia. The characterization of the aroma of Camu-Camu was recently reported, and limonene, (*E*)-caryophyllene, a-pinene, and isoamyl acetate were the compounds that most contributed to the fruity, herbal, citrus, and woody notes of the *M. dubia* fruit, also collected in Caquetá, Colombia [29]. The essential oil from *M. dubia* leaves sampled in Belém, Brazil exhibited α-pinene, (*E*)-caryophyllene, and caryophyllene oxide as its primary constituents [30].

### 3.4. Psidium guajava L.–Myrtaceae

**Botanical description:** It is a small tree, ranging in height from 10–12 m. The stems are irregular, tortuous, very branched, and light green, with quadrangular branches. The bark is thin, smooth, and greenish-brown, and exfoliates frequently. The leaves are simple, opposite in orientation, and have a short petiole. The texture of the leaves is sub-coriaceous, and they are elliptical in shape, 5–15 cm long, and 4–6 cm wide, with an obtuse, acute, or sub-acuminate apex, and an obtuse-rounded base. The lateral ribs are parallel, conspicuous, and straight. The flowers are axillary, solitary, and have a tubular-swollen hypanthium and thick greenish-white sepals. Each flower has four to five white petals, which are rounded and very deciduous, and there are numerous white stamens. The ovary is positioned inferiorly. The fruit is a rounded, ovoid, or pyriform berry of varying size, with greenish or yellow skin and numerous seeds. It is fleshy and edible [5] (Figure 8). Fruiting in two periods, from April to June/July and from November to January/February, the Goiaba is much appreciated in its natural state, with its sweet, aromatic pulp. Its primary uses are in sweets, jams, jellies, juices, and ice creams.

**Synonimy:** *Guajava pumila* (Vahl) Kuntze, *G. pyrifera* (L.) Kuntze, *Myrtus guajava* (L.) Kuntze, *Psidium angustifolium* Lam., *P. aromaticum* Blanco, *P. fragrans* Macfad., *P. guajava* L. var. guajava, *P. guayava* Radii, *P. pyriferum* L., and *Syzigium ellipticum* K. Schum. & Lauterb., among others [31].

**Geographic distribution:** It is a fruit of pre-Columbian culture, originating from Mexico to Brazil, currently cultivated in almost all New and Old-World tropical countries. See Table 4 concerning the volatile constituents identified in *Psidium guajava*.

Oxygenated sesquiterpenes (44.0%) and fatty acid derivatives (41.6%) predominated in the volatile concentrate of *P. guajava*, followed by sesquiterpenes hydrocarbons (7.4%). The main constituents were (2*E*)-hexenal (21.7%), hexenal (15.4%), caryophylla-4(12),8(13)-dien-5-β-ol (10.5%), caryophyllene oxide (9.2%), pogostol (8.3%), muurola-4,10(14)-dien-1-β-ol (4.8%), (*E*)-caryophyllene (4.1%), and (*Z*)-β-ocimene (2.6%), comprising 76.6% of the volatile concentrate (see Figure 9).

Mahattanatawee and co-workers (2005) [32] identified hexanal and (*E*)-caryophyllene as the major constituents of the volatile concentrate of Goiaba fruit sampled in Florida, USA. Also, Chen, Sheu, and Wu (2006) [33] highlighted (*E*)-caryophyllene, globulol, α-pinene, 1,8-cineole, hexanal, and ethyl hexanoate as significant components in the volatile concentrate of Goiaba fruit collected in Linnei, Taiwan. The odor-active compounds of a Goiaba specimen sampled in Alquizar, Cuba, comprised (*E*)-caryophyllene, hexanal, and 1-hexanol as the principal constituents [34]. In Brazil, the behavior of Goiaba fruit volatile compounds was found to change throughout the maturation stages: in immature fruits, the aldehydes (*E*)-2-hexenal and (*Z*)-3-hexenal predominated, and in mature fruits, the esters (*Z*)-3-hexenyl acetate and (*E*)-3-hexenyl acetate, and the sesquiterpenes (*E)*-caryophyllene, α-humulene, and β-bisabolene were the primary constituents [35]. The major constituents of the essential oil of leaves and fruits from a specimen of Goiaba sampled in Cairo, Egypt, were (*E*)-caryophyllene and limonene for the fruit, and (*E*)-caryophyllene and selin-7(11)-en-4α-ol for the leaves [36].

### 3.5. Psidium guineense Sw.–Myrtaceae

**Botanical description:** It is a species of variable size, ranging in height from 0.7 to 6.0 m. The leaves are elliptical or obovate, 8–15 cm long and 4–7 cm wide, with an obtuse or rounded apex and base. The underside surface of the leaf is more hairy, and the lateral veins exist in 8–10 pairs. The inflorescences are isolated flowers or small axillary dichasia containing up to three flowers. The flowers comprise a white corolla with free shell-shaped petals facing downwards, and around 200 stamens. The fruit is a yellowish-white globose berry, about 4 cm in diameter, containing numerous 2–3 mm seeds, hard test, and a creamy-white pulp, which is quite acidic [6] (Figure 10). It flowers from June to December, and fruits from October to March. The fruits are naturally consumed in soft drinks, ice cream, sweets, and liqueur.

**Synonimy:** *Campomanesia multiflora* (Cambess.) O. Berg, *C. tomentosa* Kunth, *Eugenia hauthalii* (Kuntze) K. Schum., *Guajava albida* (Cambess.) Kuntze, *G. guineensis* (Sw.) Kuntze, *G. multiflora* (Cambess.) Kuntze, *Mosiera guinensis* (Sw.) Bisse, *Myrtus guineensis* (Sw.) Kuntze, *Psidium albidum* Cambess., *P. araca* Raddi, *P. guyanense* Pers., and *Psidium multiflorum* Cambess., among others [37].

**Geographic distribution:** Araçá occurs in regions ranging from Mexico to the West Indies, passing through Brazil and reaching Argentina. The species has an African name due to a mistake by Swartz, who assumed it was introduced to the Antilles from Africa. Araçá is cultivated, and also occurs spontaneously, throughout the Amazon region in open areas, fields, and pastures. See Table 5 concerning the volatile constituents identified in *Psidium guineense*.

In the volatile concentrate of *P. guineense*, monoterpene hydrocarbons (36.4%), fatty acid derivatives (29.8%), oxygenated sesquiterpenes (18.9%), and sesquiterpene hydrocarbons (12.1%) predominated, complemented by a minor content of benzenoids/phenylpropanoids (1.1%) and oxygenated monoterpenes (0.4%). The primary constituents of Araçá were limonene (25.2%), ethyl butanoate (12.1%), *epi*-β-bisabolol (9.8%), α-pinene (9.2%), and ethyl hexanoate (5.9%), comprising 62.2% of its volatile concentrate (see Figure 11).

The volatiles ethyl butyrate, ethyl hexanoate, (E)-caryophyllene, and selin-11-en-4-α-ol were previously identified in Araçá fruits occurring in Colombia [38]. Furthermore, the main constituents of the fruits and leaves of an Araçá specimen collected in Hidrolândia, Goiás, Brazil, were also reported, including (2*Z*,6*E*)-farnesol, α-copaene, δ-cadinene, γ-himachalene, and cubenol in the fruits, and (2*Z*,6*E*)-farnesol, α-copaene, muurola-4,10(14)-dien-1-β-ol, and *epi*-α-cadinol in leaves [39]. Volatile compounds isolated from Araçá leaves from various locations have also been reported, with β-bisabolene and α-pinene forming the main constituents of a specimen sampled in Tempe, AZ, USA [40], high levels of spathulenol being found in leaf samples collected in Dourados, Mato Grosso do Sul, Brazil [41], and limonene, α-pinene, β-bisabolol, *epi*-α-bisabolol, *epi*-β-bisabolol, β-bisabolene, α-copaene, and (*E*)-caryophyllene appearing in specimens collected in the Amazon region of Brazil [42,43]. A review of essential oils from the leaves of *Psidium* species, emphasizing the description of monoterpenes and sesquiterpenes from *P. guineense*, was recently reported [44].

### 3.6. Fruit Scent: Chemistry and Ecological Function

Like other plant parts, fruits are also composed of secondary metabolites. These related compounds act ecologically, attracting frugivorous and seed-dispersing small animals and repelling other so-called fruit antagonists. It has been said that secondary metabolites in fruits act mainly as defensive agents for the plant. The discussion about the defense of fruits by secondary metabolites has attributed this phenomenon to molecules with higher molecular weights and non-volatile characteristics. On the other hand, less attention has been paid to volatile organic compounds and lighter, odorous hydrophobic constituents. The volatile organic compounds not only play a role in the defense of fruits, but are also responsible for their aroma and attractiveness to human consumers [45].

Fruit aroma is a significant contributor to fruit quality. In the wild, the aroma of volatile organic compounds released from fruits influences herbivore behavior. It attracts animal dispersers, such as fruit bats, that recognize ripe and non-ripe fruits based on the emitted volatiles. Additionally, volatile organic compounds from fruits have biological activities against bacteria and fungi. For example, volatiles extracted from citrus species exhibit significant antifungal and antibacterial activities against pathogenic strains [46].

Fruits are generally classified into berries, melons, citrus fruits, drupes (fruits with stones), pomes (apple and pear types), and tropical fruits, as in the present case of *Myrtaceous* species. Most fruits release a wide range of volatile organic compounds, which determine the profile of their aromas and which, in general, are fatty acid derivatives (esters, ketones, aldehydes, lactones, and alcohols), terpenoids (mono- and sesquiterpenes, benzenoids, and phenylpropanoids (aromatic compounds). Each species of fruit has a characteristic aroma based on the mixture of its volatile organic compounds [9].

Many factors regulate the aromas emitted by fruits, while the genotype of the fruit influences the flavor. The final fruit flavor profile is affected by environmental conditions, such as climate, sunlight, soil, fruit ripening, harvesting time, and post-harvesting processes. For example, environmental stresses (high temperature and drought) influence the metabolism of fruit and the aromatic compound content [47]. The volatile organic compound profiles of fruits change according to the maturation stage. Terpenoids dominate the aroma profile in some fruits, such as apples, apricots, and peaches, during ripening, while in grapes, some phenylpropanoids increase with maturation. Furthermore, fatty acid and amino acid-related compounds increase during the maturation of apples and apricots. Therefore, maturation is vital for the emission of volatile organic compounds in fruits and affects commercial production [46].

As seen, fruit aromas serve as a signal to their pollinators or eaters. However, most horticultural varieties and cultivars have been selected according to human preference. Identifying volatile organic compounds relevant to human sensory preferences is essential to meet consumer demand for fruits. Furthermore, biotechnological modification of the aromatic characteristics of fruits or the engineering of synthesis pathways in microbial cell factories could increase the production of their aromatic metabolites for commercial exploitation [48].

### 3.7. Multivariate Statistical Analysis

The variability of Myrtaceae fruit volatile constituents was evaluated using multivariate statistical analyses (PCA, principal component analysis; and HCA, hierarchical cluster analysis) based on their classes of compounds. The percentage values of monoterpene hydrocarbons (MH), oxygenated monoterpenes (OM), sesquiterpene hydrocarbons (SH), oxygenated sesquiterpenes (OS), fatty acid derivatives (FA), and benzenoids/phenylpropanoids (B/P) were obtained based on the GC–MS analyses of their volatile constituents. The data of compound classes from Table 1, Table 2, Table 3, Table 4 and Table 5 was used as variables (see Table 6).

The HCA analysis (Figure 12) showed a heterogeneous formation of five groups, with a similarity of only 55.53% between species. The first group was composed of *Eugenia stipitata* (Esti, I); the second group of *Myrciaria dubia* (Mdub, II); the third group of *Eugenia uniflora* (Euni, III); the fourth group of *Psidium guajava* (Pgua, IV); and the fifth group of *Psidium guineense* (Pgui, V), showing a statistical differentiation between these and, therefore, a significant chemical variability (among the classes of compounds analyzed) in the species of analyzed fruits. This chemical variability is also associated with the aromas and flavors of the fruits, which are very distinct.

The analysis of chemical variability was also evaluated via principal component analysis (PCA), which represented 83.9% of the data (Figure 13), in which PC1 explained 54.5% of the data and showed a negative correlation with oxygenated monoterpenes (OM, λ = −0.389) and monoterpene hydrocarbons (MH, λ = −0.564), and a positive correlation with oxygenated sesquiterpenes (OS, λ = 0.584), sesquiterpene hydrocarbons (SH, λ = 0.264), and fatty acid derivatives, plus benzenoids/phenylpropanoids (FA-B/P, λ = 0.346). PC2 justified 29.4% of the data and showed a positive correlation with monoterpene hydrocarbons (MH, λ = 0.002) and fatty acid derivatives plus benzenoids/phenylpropanoids (FA-B/P, λ = 0.670), and a negative correlation with oxygenated monoterpenes (OM, λ = −0.060), sesquiterpene hydrocarbons (SH, λ = −0.733), and oxygenated sesquiterpenes (OS, λ = −0.104).

Like HCA, the PCA analysis confirmed the formation of five different groups. Therefore, the chemical variability between its main classes of compounds can be justified as follows. The fruits of *Eugenia stipitata* (Esti) and *Myrciaria dubia* (Mdub) were characterized by the presence of monoterpene hydrocarbons (Esti, 28.5%; Mdub, 79.6%) and oxygenated monoterpenes (Esti, 25.5%; Mdub, 11.5%). The fruit of *Eugenia uniflora* (Euni) was distinguished by the existence of sesquiterpene hydrocarbons (41.3%). The fruit of *Psidium guajava* (Pgua) was characterized by the presence of oxygenated sesquiterpenes (43.8%) and fatty acid derivatives (41.6%). The fruit of *Psidium guineense* (Pgui) was distinguished by the existence of monoterpene hydrocarbons (36.4%) and fatty acid derivatives (29.8%).

Also based on the PCA and HCA studies, it was observed that there was no significant statistical grouping between the analyzed samples whose chemical profiles were characterized by the volatile constituents α-pinene (17.4%), citronellyl butanoate (15.6%), pogostol (13.5%), and α-terpineol (9.6%) in *Eugenia stipitata*; curzerene (30.5%), germacrone (15.4%), atractylone (13.1%), and (*E*)-β-ocimene (11.1%) in *Eugenia uniflora*; α-pinene (55.8%), (*E*)-β-ocimene (13.1%), and α-terpineol (10%) in *Myrciaria dubia*; (2*E*)-hexenal (21.7%), hexanal (15.4%), caryophylla-4(12),8(13)-dien-5-β-ol (10.5%), and caryophyllene oxide (9.2%) in *Psidium guajava*; and limonene (25.2%), ethyl butanoate (12.1%), epi-β-bisabolol (9.8%), and α-pinene (9.2%) in *Psidium guineense*.

## 4. Conclusions

The present study contributed to improved knowledge of the chemotaxonomy of Myrtaceae fruit species, of which there are few reports in the existing literature. The main classes of compounds in the studied species were determined as follows: in *Eugenia stipitata*, monoterpene hydrocarbons, oxygenated monoterpenes, sesquiterpene hydrocarbons, oxygenated sesquiterpenes, and fatty acid derivatives were highly represented; in *E. uniflora*, there existed an absence of oxygenated monoterpenes and fatty acid derivatives; in *Myrciaria dubia*, there were only monoterpene hydrocarbons and oxygenated monoterpenes; in *Psidium guajava*, sesquiterpene hydrocarbons, oxygenated sesquiterpenes, and fatty acid derivatives predominated; and in *P. guineense*, we observed an absence of oxygenated monoterpenes. Therefore, these findings contribute to a better understanding of the chemical profiles of Myrtaceae fruit species.

## Figures and Tables

**Figure 1 foods-13-01490-f001:**
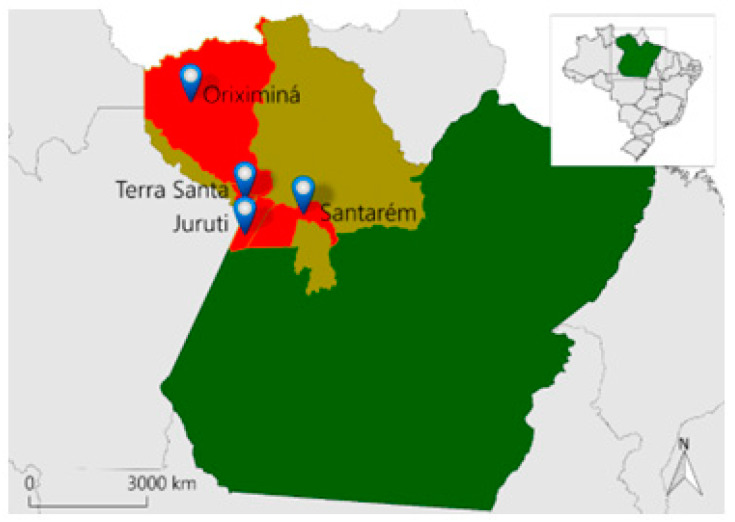
Location of fruit collection areas in the Brazilian Amazon.

**Figure 2 foods-13-01490-f002:**
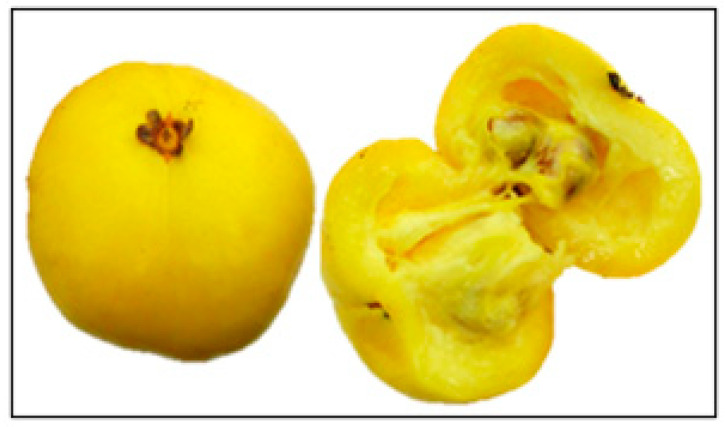
*Eugenia stipitata* fruits—trivial name Araçá-boi.

**Figure 3 foods-13-01490-f003:**
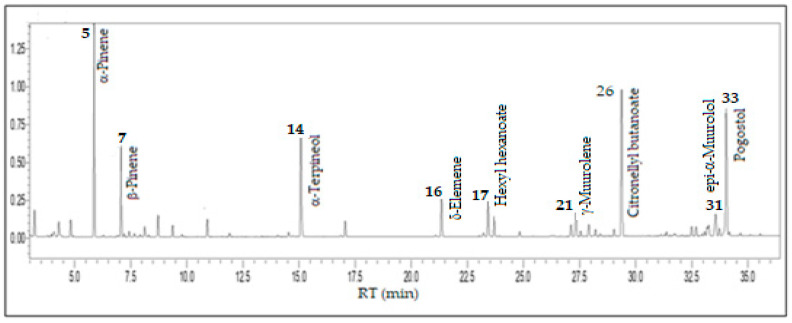
Ion-chromatogram of the *Eugentipitateata* fruit volatile concentrate.

**Figure 4 foods-13-01490-f004:**
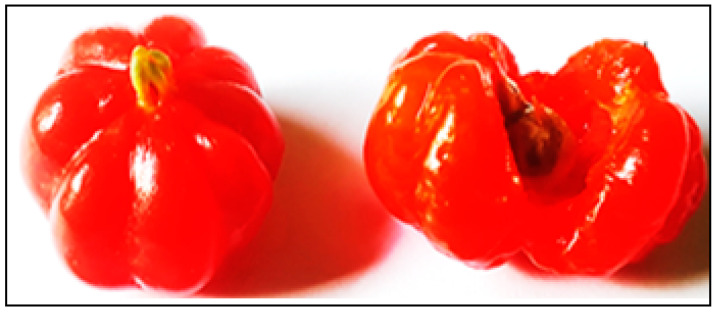
*Eugenia uniflora* fruits—trivial names Ginja and Pitanga.

**Figure 5 foods-13-01490-f005:**
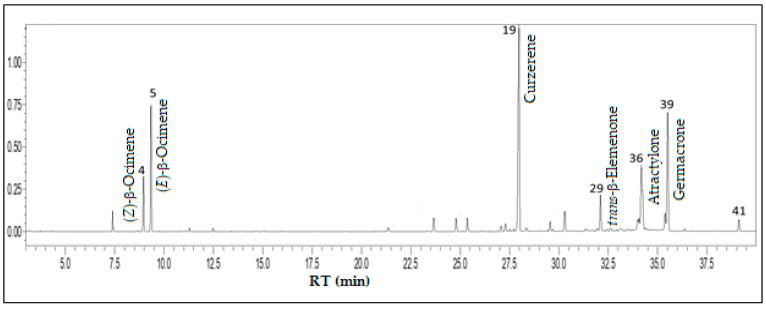
Ion-chromatogram of the *Eugenia uniflora* fruit volatile concentrate.

**Figure 6 foods-13-01490-f006:**
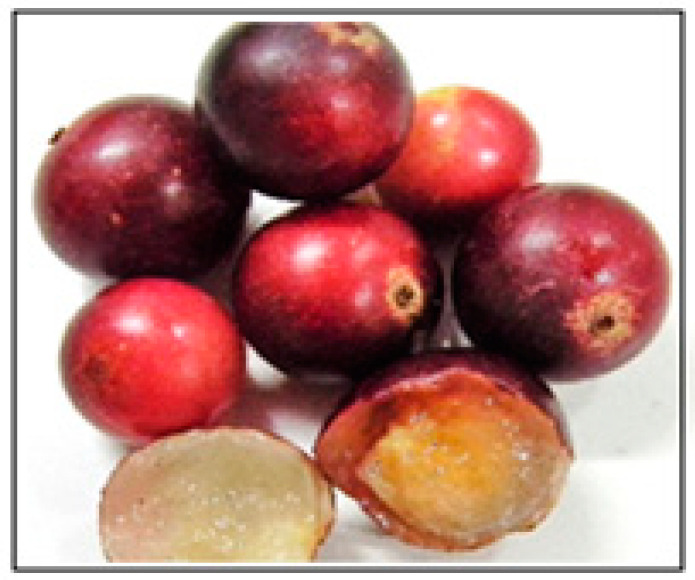
*Myrciaria dubia* fruits—common name Camu-Camu.

**Figure 7 foods-13-01490-f007:**
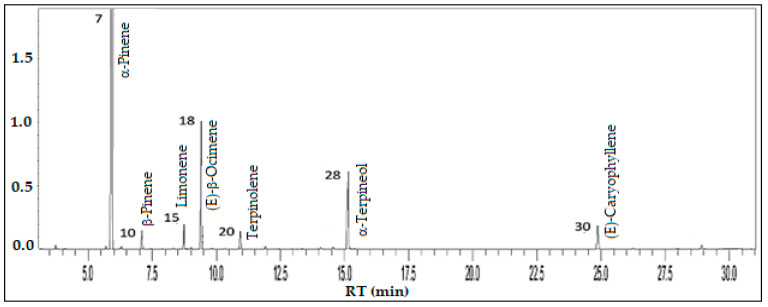
Ion-chromatogram of the *Myrciaria dubia* fruit volatile concentrate.

**Figure 8 foods-13-01490-f008:**
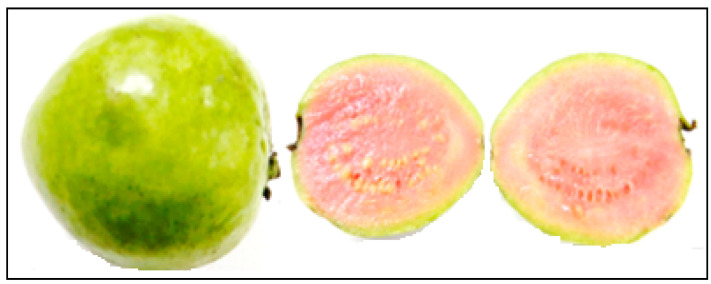
*Psidium guajava* fruits—common name Goiaba.

**Figure 9 foods-13-01490-f009:**
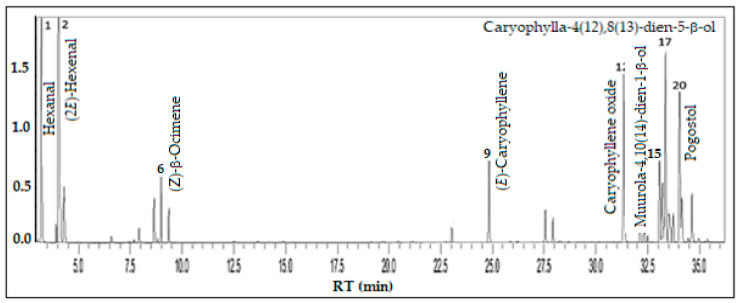
Ion-chromatogram of the *Psidium guajava* fruit volatile concentrate.

**Figure 10 foods-13-01490-f010:**
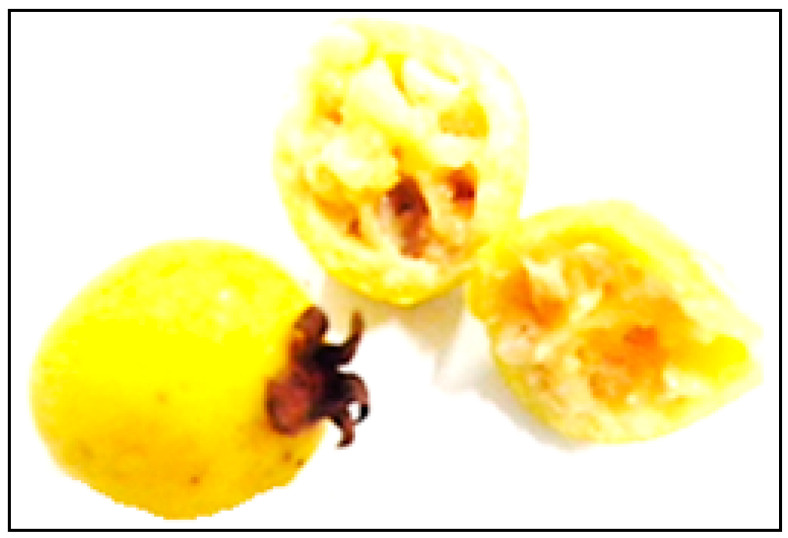
*Psidium guineense* fruits—Araçá.

**Figure 11 foods-13-01490-f011:**
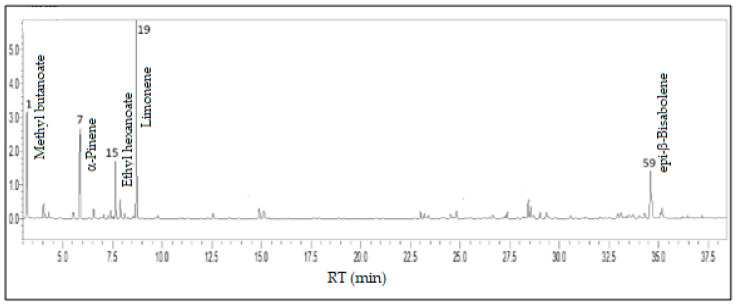
Ion-chromatogram of the *Psidium guineense* fruit volatile concentrate.

**Figure 12 foods-13-01490-f012:**
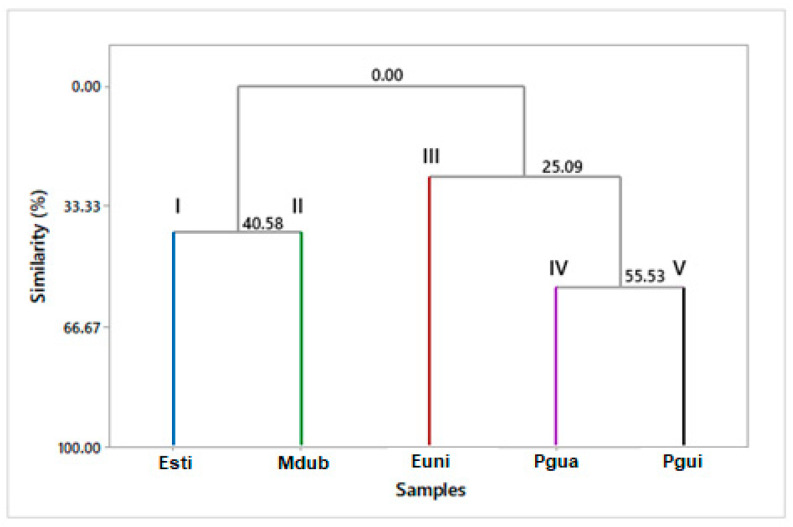
Hierarchical cluster analysis (HCA) of the Myrtaceae fruit volatile concentrates, based on their classes of compounds.

**Figure 13 foods-13-01490-f013:**
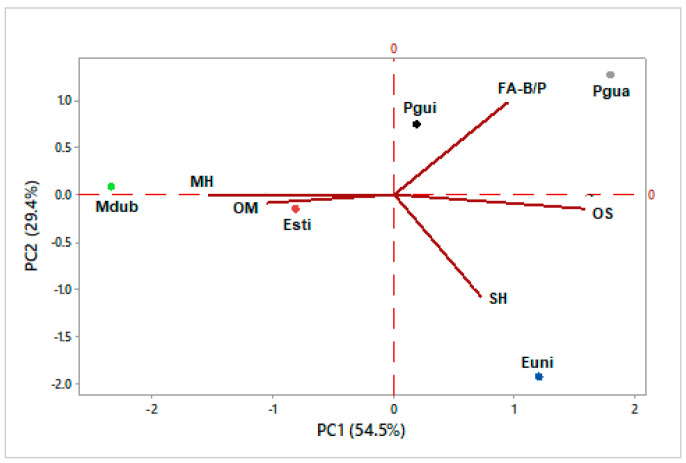
Principal component analysis (PCA) of the Myrtaceae fruit volatile concentrates, based on their classes of compounds.

**Table 1 foods-13-01490-t001:** Constituents identified in the volatile concentrate of *Eugenia stipitata* fruits.

Constituents	RI_Cal_	RI_Lit_	%
Ethyl butanoate	799	802 ^a^	2.1
(3*Z*)-Hexenol	848	850 ^a^	0.6
*n*-Hexanol	860	863 ^a^	1.1
2-Heptanol	894	891 ^b^	1.2
**α-Pinene**	**932**	**932 ^a^**	**17.4**
Hexanoic acid (Caproic acid)	970	967 ^a^	0.1
**β-Pinene**	**976**	**974 ^a^**	**6.8**
Myrcene	989	988 ^a^	0.4
Hexyl acetate	1011	1007 ^a^	0.8
Limonene	1027	1024 ^a^	1.7
(*E*)-β-Ocimene	1045	1044 ^a^	0.8
p-Mentha-2,4(8)-diene	1087	1085 ^a^	1.4
Terpinen-4-ol	1176	1174 ^a^	0.3
**α-Terpineol**	**1189**	**1186 ^a^**	**9.6**
n-Hexyl 2-methyl butanoate	1236	1233 ^a^	1.4
δ-Elemene	1336	1335	4.1
Hexyl hexanoate	1385	1382 ^a^	3.5
β-Elemene	1392	1389 ^a^	1.8
(*E*)-Caryophyllene	1419	1417 ^a^	0.4
β-Chamigrene	1475	1476 ^a^	1.2
γ-Muurolene	1481	1478 ^a^	2.6
Germacrene D	1486	1484 ^a^	0.5
β-Selinene	1492	1489 ^a^	1.3
(Z)-α-Bisabolene	1502	1506 ^a^	0.5
δ-Cadinene	1523	1522 ^a^	0.7
**Citronellyl butanoate**	**1532**	**1530 ^a^**	**15.6**
Caryophyllene oxide	1583	1582 ^a^	0.4
Junenol	1618	1618 ^a^	1.9
1-*epi*-Cubenol	1628	1627 ^a^	0.3
γ-Eudesmol	1631	1630 ^a^	0.8
*epi*-α-Muurolol	1642	1640 ^a^	3.2
Cubenol	1646	1645 ^a^	0.8
**Pogostol**	**1654**	**1651 ^a^**	**13.5**
Monoterpene hydrocarbons Oxygenated monoterpenes Sesquiterpene hydrocarbons Oxygenated sesquiterpenes Fatty acid derivatives			28.5
			25.5
			13.1
			20.9
			10.8
Total (%)			98.8

RI_Cal_ = Calculated Retention Index; RI_Lit_ = Literature Retention Index; ^a^ Adams, 2007 [11]; ^b^ Mondello, 2011 [12]; Bold = Main constituents. Unidentified minor constituents were 1.2%.

**Table 2 foods-13-01490-t002:** Constituents identified in the volatile concentrate of *Eugenia uniflora* fruits.

Constituents	RI_Cal_	RI_Lit_	%
*n*-Octane	797	800 ^a^	0.1
Myrcene	989	988 ^a^	1.6
Limonene	1027	1024 ^a^	0.1
**(Z)-β-Ocimene**	**1034**	**1032 ^a^**	**4.6**
**(*E*)-β-Ocimene**	**1045**	**1044 ^a^**	**11.1**
Linalool	1098	1095 ^a^	0.3
*allo*-Ocimene	1127	1128 ^a^	0.1
δ-Terpineol	1161	1162 ^a^	0.3
α-Terpineol	1189	1186 ^a^	0.1
δ-Elemene	1337	1335 ^a^	0.4
β-Elemene	1391	1389 ^a^	1.5
(*E*)-Caryophyllene	1419	1417 ^a^	1.5
γ-Elemene	1433	1434 ^a^	1.5
α-Humulene	1453	1452 ^a^	0.1
β-Chamigrene	1475	1476 ^a^	0.6
γ-Muurolene	1480	1478 ^a^	0.9
β-Selinene	1485	1489 ^a^	0.2
δ-Selinene	1490	1492 ^a^	0.2
**Curzerene**	**1497**	**1499 ^a^**	**30.5**
*cis*-α-Bisabolene	1509	1506 ^a^	0.1
δ-Cadinene	1523	1522 ^a^	0.1
γ-Cuprenene	1534	1532 ^a^	0.1
α-Cadinene	1537	1537 ^a^	1.1
Selina-4(15),7(11)-diene	1541	1540 ^a^	0.2
Germacrene B	1556	1557 ^a^	2.3
Spathulenol	1576	1577 ^a^	0.4
Cubeban-11-ol	1593	1595 ^a^	0.2
*cis*-β-Elemenone	1589	1589 ^a^	0.3
***trans*-β-Elemenone**	**1603**	**1601 ^a^**	**4.1**
1,10-di-*epi*-Cubenol	1617	1618 ^a^	0.4
10-*epi*-γ-Eudesmol	1625	1622 ^a^	0.1
γ-Eudesmol	1631	1630 ^a^	0.5
*epi*-α-Muurolol	1641	1640 ^a^	0.2
Pogostol	1654	1651 ^a^	1.2
**Atractylone**	**1659**	**1657 ^a^**	**13.1**
Selin-11-en-4-α-ol	1662	1658 ^a^	0.2
**Germacrone**	**1692**	**1693 ^a^**	**15.4**
Zizanal	1694	1697 ^a^	2.1
Maiurone	1709	1709 ^a^	0.3
γ-Eudesmol acetate	1780	1783 ^a^	1.3
Monoterpene hydrocarbons Oxygenated monoterpenes Sesquiterpene hydrocarbons Oxygenated sesquiterpenes Fatty acid derivatives			17.5
			0.7
			41.3
			39.8
			0.1
Total (%)			99.4

RI_Cal_ = calculated retention index; RI_Lit_ = literature retention undex; ^a^ Adams, 2007 [11]; Bold = main constituents. Unidentified minor constituents comprised 0.6%.

**Table 3 foods-13-01490-t003:** Constituents identified in the volatile concentrate of *Myrciaria dubia* fruits.

Constituents	RI_Cal_	RI_Lit_	%
2,4-Dimethyl-3-pentanone	795	788 ^a^	0.1
(3*Z*)-Hexenal	798	797 ^a^	0.1
Furfural	827	828 ^a^	0.3
(2*E*)-Hexenal	847	846 ^a^	0.1
(3*Z*)-Hexenol	851	850 ^a^	0.1
α-Thujene	925	924 ^a^	0.3
**α-Pinene**	**934**	**932 ^a^**	**55.8**
α-Fenchene	946	945 ^a^	0.1
Camphene	947	946 ^a^	0.2
**β-Pinene**	**976**	**974 ^a^**	**2.6**
Myrcene	989	988 ^a^	0.1
α-Phellandrene	1005	1002 ^a^	0.1
α-Terpinene	1016	1014 ^a^	0.1
*p*-Cymene	1023	1020 ^a^	0.1
**Limonene**	**1027**	**1024 ^a^**	**3.7**
1,8-Cineole	1030	1026 ^a^	0.1
(*Z*)-β-Ocimene	1035	1032 ^a^	0.2
**(*E*)-β-Ocimene**	**1046**	**1044 ^a^**	**13.1**
γ-Terpinene	1057	1054 ^a^	0.3
**Terpinolene**	**1087**	**1086 ^a^**	**2.9**
*endo*-Fenchol	1112	1114 ^a^	0.3
α-Campholenal	1125	1122 ^a^	0.1
*trans*-Pinocarveol	1138	1135 ^a^	0.1
*cis*-β-Terpineol	1143	1140 ^a^	0.1
Camphene hydrate	1147	1145 ^a^	0.1
Borneol	1164	1165 ^a^	0.3
Terpinen-4-ol	1176	1174 ^a^	0.3
**α-Terpineol**	**1190**	**1186 ^a^**	**10.0**
γ-Terpineol	1196	1199 ^a^	0.1
**(*E*)-Caryophyllene**	**1419**	**1417 ^a^**	**4.2**
γ-Elemene	1433	1434 ^a^	0.1
α-Humulene	1453	1452 ^a^	0.2
Bicyclogermacrene	1496	1497 ^a^	0.1
δ-Amorphene	1508	1511 ^a^	0.1
δ-Cadinene	1523	1522 ^a^	0.1
Germacrene B	1556	1559 ^a^	0.4
Monoterpene hydrocarbons			79.6
Oxygenated monoterpenes			11.5
Sesquiterpene hydrocarbons			5.2
Fatty acid derivatives			0.7
Total (%)			97.0

RI_Cal_ = calculated retention index; RI_Lit_ = literature retention index; ^a^ Adams, 2007 [11]; Bold = main constituents. Unidentified minor constituents comprised 3.0%.

**Table 4 foods-13-01490-t004:** Constituents identified in the volatile concentrate of *Psidium guajava* fruits.

Constituents	RI_Cal_	RI_Lit_	%
**Hexanal**	**800**	**801 ^a^**	**15.4**
**(2*E*)-Hexenal**	**847**	**846 ^a^**	**21.7**
*n*-Hexanol	862	863 ^a^	2.2
(3*Z*)-Hexenyl acetate	1006	1004 ^a^	0.5
2-Ethylhexanol	1026	1030 ^a^	1.8
**(*Z*)-β-Ocimene**	**1036**	**1032 ^a^**	**2.6**
(*E*)-β-Ocimene	1046	1044 ^a^	1.3
α-Copaene	1376	1374 ^a^	0.6
**(*E*)-Caryophyllene**	**1420**	**1417 ^a^**	**4.1**
β-Selinene	1487	1489 ^a^	1.6
α-Selinene	1496	1498 ^a^	1.1
**Caryophyllene oxide**	**1583**	**1582 ^a^**	**9.2**
Ledol	1604	1602 ^a^	0.9
Humulene epoxide II	1609	1608 ^a^	0.6
**Muurola-4,10(14)-dien-1-β-ol**	**1629**	**1630 ^a^**	**4.8**
Caryophylla-4(12),8(13)-dien-5-α-ol	1636	1639 ^a^	3.3
**Caryophylla-4(12),8(13)-dien-5-β-ol**	**1637**	**1639 ^a^**	**10.5**
α-Muurolol	1641	1644 ^a^	1.4
β-Eudesmol	1645	1649 ^a^	2.4
**Pogostol**	**1648**	**1651 ^a^**	**8.3**
14-hydroxy-9-*epi*-(*E*)-Caryophyllene	1664	1668 ^a^	2.4
Monoterpene hydrocarbons			3.9
Sesquiterpene hydrocarbons			7.4
Oxygenated sesquiterpenes			43.8
Fatty acid derivatives			41.6
Total (%)			96.7

RI_Cal_ = calculated retention index; RI_Lit_ = literature retention index; ^a^ Adams, 2007 [13]; Bold = main constituents. Unidentified minor constituents comprised 3.1%.

**Table 5 foods-13-01490-t005:** Constituents identified in the volatile concentrate of *Psidium guineense* fruits.

Constituents	RI_Cal_	RI_Lit_	%
**Ethyl butanoate**	**799**	**802 ^a^**	**12.1**
Butyl acetate	806	807 ^a^	0.2
(2E)-Hexenal	846	846 ^a^	2.6
*n*-Hexanol	860	863 ^a^	0.8
Isopentyl acetate	870	869 ^a^	0.1
Methyl hexanoate	920	921 ^a^	0.6
**α-Pinene**	**932**	**932 ^a^**	**9.2**
Camphene	947	946 ^a^	0.1
Benzaldehyde	955	952 ^a^	1.1
Hexanoic acid	970	967 ^a^	0.1
β-Pinene	976	974 ^a^	0.4
6-methyl-5-Hepten-2-one	984	981 ^a^	0.4
Myrcene	989	988 ^a^	0.9
Butyl butanoate	994	993 ^a^	0.2
**Ethyl hexanoate**	**998**	**997 ^a^**	**5.9**
(3*Z*)-Hexenyl acetate	1005	1004 ^a^	2.3
Hexyl acetate	1011	1007 ^a^	0.6
*p*-Cymene	1023	1020 ^a^	0.3
**Limonene**	**1027**	**1024 ^a^**	**25.2**
1,8-Cineole	1030	1026 ^a^	0.2
γ-Terpinene	1057	1054 ^a^	0.3
Methyl octanoate	1123	1123 ^a^	0.1
(3Z)-Hexenyl butanoate	1185	1184 ^a^	1.2
Hexyl butanoate	1191	1191 ^a^	1.4
Ethyl octanoate	1196	1196 ^a^	0.1
α-Copaene	1376	1374	0.9
(3*Z*)-Hexenyl hexanoate	1380	1378 ^a^	0.7
Geranyl acetate	1383	1379 ^a^	0.2
Hexyl hexanoate	1385	1382 ^a^	0.4
*iso*-Italicene	1403	1401 ^a^	0.1
Acora-3,7(14)-diene	1408	1407 ^a^	0.2
α-Cedrene	1412	1410 ^a^	0.6
(*E*)-Caryophyllene	1420	1417 ^a^	1.1
β-Santalene	1460	1457 ^a^	0.2
α-Acoradiene	1464	1464 ^a^	0.4
10-*epi*-β-Acoradiene	1479	1474 ^a^	0.4
*Ar*-Curcumene	1482	1479 ^a^	0.9
α-Zingiberene	1495	1493 ^a^	0.1
(*Z*)-α-Bisabolene	1502	1503 ^a^	0.2
β-Bisabolene	1508	1506 ^a^	2.7
α-Bulnesene	1512	1509 ^a^	0.7
β-Curcumene	1515	1514 ^a^	1.8
δ-Cadinene	1524	1522 ^a^	0.9
(*E*)-γ-Bisabolene	1532	1529 ^a^	0.9
(*E*)-Nerolidol	1563	1561 ^a^	0.4
Caryophyllene oxide	1583	1582 ^a^	0.2
Cedrol	1601	1600 ^a^	0.1
10-*epi*-γ-Eudesmol	1625	1622 ^a^	0.7
α-Acorenol	1630	1632 ^a^	1.1
Gossonorol	1637	1636 ^a^	0.2
epi-α-Cadinol	1641	1638 ^a^	0.6
Hinesol	1644	1640 ^a^	0.3
α-Muurolol	1646	1644 ^a^	0.6
α-Cadinol	1654	1652 ^a^	0.5
14-hydroxy-(*Z*)-Caryophyllene	1664	1666 ^a^	0.7
** *epi* ** **-β-Bisabolol**	**1670**	**1670 ^a^**	**9.8**
*epi*-α-Bisabolol	1683	1683 ^a^	0.8
α-Bisabolol	1685	1685 ^a^	1.8
(2*E*,6*Z*)-Farnesal	1714	1713 ^a^	0.3
(2*Z*,6*E*)-Farnesol	1721	1722 ^a^	0.4
(2*E*-6*E*)-Farnesal	1741	1740 ^a^	0.4
Monoterpene hydrocarbons			36.4
Oxygenated monoterpenes			0.4
Sesquiterpene hydrocarbons			12.1
Oxygenated sesquiterpenes			18.9
Benzenoids/Phenylpropanoids			1.1
Fatty acid derivatives			29.8
Total (%)			98.7

RI_Cal_ = calculated retention index; RI_Lit_ = literature retention index; ^a^ Adams, 2007 [11]; Bold = main constituents. Unidentified minor constituents comprised 1.3%.

**Table 6 foods-13-01490-t006:** Classes of compounds identified in the Myrtaceae fruits and used in the multivariate statistical analyses.

Classes of Compounds (%)	Esti	Euni	Mdub	Pgua	Pgui
Monoterpene hydrocarbons (MH)	28.5	17.5	79.6	3.9	36.4
Oxygenated monoterpenes (OM)	25.5	0.7	11.5	-	0.4
Sesquiterpene hydrocarbons (SH)	13.1	41.3	5.2	7.4	12.1
Oxygenated sesquiterpenes (OS)	20.9	39.8	-	43.8	18.9
Fatty acid derivatives (FA)	10.8	0.1	0.7	41.6	29.8
Benzenoids/Phenylpropanoids (B/P)	-	-	-	-	1.1
Total (%)	98.8	99.4	97.0	96.7	98.7

Esti = Eugenia stipitata; Euni = Eugenia uniflora; Mdub = Myrciaria dubia; Pgua = Psidium guajava; Pgui = Psidium guineense.

## Data Availability

The original contributions presented in the study are included in the article, further inquiries can be directed to the corresponding author.

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
