# Peer review of "Volatile Constituents of Some Myrtaceous Edible and Medicinal Fruits from the Brazilian Amazon"

_foods, 2024, doi:10.3390/foods13101490_

Round 1
Reviewer 1 Report
Comments and Suggestions for Authors
The manuscript describes several GC-MS analysis of essential oils obtained by simultaneous distillation and extraction (SDE-Likens & Nickerson-type apparatus) from fruit pulps from Eugenia stipitata (Araçá-boi), Eugenia uniflora (Ginja), Myrciaria dubia (Camu-Camu), Psidium guajava (Guava), and Psidium guineense (Araçá), common edible fruit species from Brazilian North region. If possible,please include, from the abstract, the fruit names in english.
Title, abstract and keywords are ok.
Figure 1 could be moved to the experimental section.
The experimental section is well-described. Indeed, the number of fruits from each species and replicates (only duplicate on extraction, as presented at line 98?) should be included. Also, the biodiversity access autorization number should be included. If the chromatogrpahic analysis, why the SDs were not presented?
It is suggested that the botanical description of the especies could be excluded, keeping only the data from fruits.
Octane was detected, but pentane was used at SDE, please clarify if there is no chance of contamination. Also, it is not clear how the oils were diluted to GCMS analysis.
References are ok, updated,with only 10% of self-citations,but some of them out of format.
Comments on the Quality of English LanguageNone.
Reviewer 2 Report
Comments and Suggestions for Authors
The manuscript under appreciation presents the analysis of volatile compounds by gas chromatography of fruits from the Brazilian Amazon (Eugenia stipitata (Araçá-boi), Eugenia uniflora (Ginja), Myrciaria dubia (Camu-Camu), Psidium guajava (Goiaba), Psidium guineense (Araçá)).
The manuscript is interesting, provides novelty, and is presented in a well-structured manner.
The introduction is sufficient, and the methods are adequately described.
The composition of volatile compounds is affected by many factors, including maturity and harvest periods. Therefore, it is strongly suggested that the authors include this information in section “2.1 Plant material”.
In the “3. Results and Discussion” section, after the presentation of the results for each fruit, the authors reported results from previous research studies (e.g. lines 167-175, 204-217, 248-257, 289-301, 334-346), however at this point the manuscript lacks critical comparison. Therefore, the authors should add a discussion that reveals the differences with the previous studies (for example: different extraction methods, maturity stages, geographical differences, etc.) to highlight the novelty of the current study.
In lines 412-418 critical discussion is needed: please provide a possible explanation for the negative and positive correlation that was observed between the classes of compounds after PCA analysis. Were these results expected?
Figures 3,5,7,9 need to be improved.
The conclusions are consistent with the evidence and arguments presented.
Reviewer 3 Report
Comments and Suggestions for Authors
Please provide a detailed description of the extraction process for the volatile concentrates from the fruit pulp.
Chromatographic analysis should be conducted at least three times for reproducibility, rather than just twice.
In Table 4, the sum of identified volatile components and unidentified minor components does not add up to 100%.
In Multivariate statistical analysis and “The HCA analysis (Figure 12) showed a heterogenous formation of five groups, with a similarity of 55.53% between the species.”
For "The HCA analysis (Figure 12) showed a heterogenous formation of five groups, with a similarity of 55.53% between the species ", in Section 3.7, the 55.53% of which lacks a proper explanation
Insufficient explanation and elaboration were provided regarding the results of PCA and HCA analyses. More details and discussion are needed to elucidate the significance of the PCA and HCA analysis results.
Comments on the Quality of English LanguageEnglish writing grammar is basically without any problems.
